# Impact of Nitrogen Addition on Physiological, Crop Total Nitrogen, Efficiencies and Agronomic Traits of the Wheat Crop under Rainfed Conditions

**Umara Qadeer [1,\*], Mukhtar Ahmed [1,2,3,\*]** **, Fayyaz-ul -Hassan [1] and Muhammad Akmal [4]**

[1] Department of Agronomy, Pir Mehr Ali Shah Arid Agriculture University, Rawalpindi 46300, Pakistan; drsahi63@gmail.com

[2] Department of Agricultural Research for Northern Sweden, Swedish University of Agricultural Sciences, Umeå 90183, Sweden

[3] Department of Biological Systems Engineering, Washington State University Pullman, Pullman, WA 99164-6120, USA

[4] Institute of Soil Science, Pir Mehr Ali Shah Aridl Agriculture University, Rawalpindi 46300, Pakistan; akmal@uaar.edu.pk

\* Correspondence: across-de-world@hotmail.com (U.Q.); mukhtar.ahmed@slu.se or ahmadmukhtar@uaar.edu.pk (M.A.)

**Abstract:** Optimizing nitrogen (N) application timings and rate can improve nutrient uptake and nutrient efficiencies in wheat, particularly under rainfed conditions. Climatic stress in the form of high temperature and drought resulted in the decreased crop physiological traits, hastened maturity and, ultimately, caused lower grain yield. The impact of N application rates as full and split dose at three diverse locations of rainfed Pothwar, Pakistan was studied through field experiments for two years (2013–14 and 2014–15). Treatments include $T_1$ = control (no fertilizer applied), full dose of N applied at the time of crop sowing, i.e., $T_2$ = 50 kg N ha$^{-1}$, $T_3$ = 100 kg N ha$^{-1}$ and $T_4$ = 150 kg N ha$^{-1}$, and split application of N at different timings at different stages of the crop, called split application of N, i.e., $T_5$: application of 50 kg N ha$^{-1}$ (15 kg N ha$^{-1}$ (sowing, BBCH (Biologische Bundesanstalt Bundessortenamt und Chemische Industrie) 0): 20 kg N ha$^{-1}$ (tillering, BBCH20): 15 kg N ha$^{-1}$ (anthesis, BBCH 60), $T_6$: application of 100 kg N ha$^{-1}$ (30 kg N ha$^{-1}$ (sowing, BBCH 0): 40 kg N ha$^{-1}$ (tillering, BBCH 20): 30 kg N ha$^{-1}$ (anthesis, BBCH 60) and $T_7$: application of 150 kg N ha$^{-1}$ (45 kg N ha$^{-1}$ (sowing, BBCH 0): 60 kg N ha$^{-1}$ (tillering, BBCH 20): 45 kg N ha$^{-1}$ (anthesis, BBCH 60). The three study sites were Islamabad (high rainfall with optimum temperature), University Research Farm (URF)-Chakwal Road, Koont (medium rainfall with moderate temperature), and Talagang (low rainfall with high temperature). Results revealed that the highest stomatal conductance (0.80 mole H$_2$O m$^{-2}$ s$^{-1}$), net photosynthetic rate (20.07 μmole CO$_2$ m$^{-2}$ s$^{-1}$), transpiration rate (9.58 mmole H$_2$O m$^{-2}$ s$^{-1}$), intercellular CO$_2$ concentration (329.25 μmole CO$_2$ mol$^{-1}$ air), SPAD values (58.86%) and proline contents (35.42 μg g$^{-1}$) were obtained from split application of N ($T_6$ = split N$_{100}$) compared to control and full dose N treatments. Among the sites, these physiological traits remained highest at Islamabad and lowest at Talagang, while between the years, the maximum values of the measured parameters were obtained during 2013–14. A similar trend was observed for crop total N, N efficiencies, and agronomic traits of the crop. The results suggested that the optimum N application rate at appropriate timings can help to harvest the real benefits of N. The split dose resulted in the maximum performance of the crop from the physiological parameters to the agronomic traits of the crop.

**Keywords:** climate; rainfed wheat; N fertilization; split and full N application; photosynthetic rate; agronomic traits

## 1. Introduction

Rainfed regions occupy 45% of the earth and are home to 38% of global production [1]. Water stress and low nitrogen (N) fertility are the main constraints lowering crop productivity of this region. These limitations are expected to increase in the future as climate change could result in a decrease of rainfall and increase of evapotranspiration in the rainfed region. Climate change in the form of rising temperatures, rainfall variability, and elevated $CO_2$ is showing significant impact on the productivity of this region. Since elevated $CO_2$ [$eCO_2$] is one of the documented global changes, it is thus showing its effects on plants in the form of reduction in stomatal conductance ($g_s$) and transpiration rate (E) and increased light use efficiency [2]. However, this [$eCO_2$] could help to increase the photosynthesis of crops by increasing the Ribulose-1,5-bisphosphate carboxylase (Rubisco) carboxylation rate and inhibition of its oxygenation [3]. It has been reported that [$eCO_2$] resulted in a 31% increase in the light-saturated leaf photosynthetic rate and 28% daily photosynthetic carbon assimilation [4]. However, stressful conditions like low N and drought resulted in decreased $g_s$ and a reduced net photosynthetic rate [4]. A similar trend of decreased $g_s$ (20%) was observed for $C_3$ and $C_4$ species. The decrease in photosynthesis is more pronounced under N-limited conditions as less N supply might limit the development of new sinks and disturbs the source-sink balance in plant growth under [$eCO_2$] [5,6].

Dry matter production in plants depends upon the process of photosynthesis. It is the process in which plants convert light energy into photoassimilates through the action of $CO_2$ and water. However, the supply of N is also one of the determinant factors for the process of photosynthesis. During the light reaction of photosynthesis, a generated electron is passed to Nicotinamide adenine dinucleotide phosphate (NADP) and helps to create NADPH by accepting hydrogen from the photolysis of water. This NADPH is then used in the $CO_2$ carboxylation process to generate photosynthate [7]. However, water and nutrient deficiency, mainly N, could lead to a decrease in $g_s$ (stomatal conductance) and intercellular $CO_2$ concentration (Ci) due to a reduced transpiration rate (E), resulting in a reduced photosynthetic $CO_2$ carboxylation rate [8–10]. This reduction in crop physiological traits could be improved by the application of N, as it affects plant adaptation to abiotic stress [11–13]. The physiological process in plants is significantly affected by N deficiency, and it is well documented that N deficiency has an impact on the photosynthetic $CO_2$ assimilation rate. It has been reported that lower levels of N lead to the $CO_2$ assimilation rate by the leaves leading to less photosynthetic yield [14–19]. Furthermore, it has been confirmed that this decrease in the photosynthetic $CO_2$ assimilation capacity is related to the supply of N, which resulted in a decreased Rubisco content and activity of an enzyme (RuBPcase) in the dark reaction of photosynthesis [20].

A positive correlation between the photosynthetic capacity of leaves and N contents has been shown in earlier work [14,21–23]. Application of N in wheat (*Triticum aestivum* L.) plants resulted in improved properties of photosynthetic pigments and increased net photosynthetic rate ($A_n$) [24,25]. N deficiency could decrease the yield of PSII (Photosystem II) and $CO_2$ assimilation of photosynthesis, resulting in the lower crop yield [26–28]. However, some earlier work depicted that N deficiency has no effect on the PSII but has an impact on the return of $CO_2$ assimilation and light-saturated rate of photosynthesis [26]. Lu and Zhang, (2000) [29] studied the effect of N deficiency on photosynthetic $CO_2$ assimilation, PSII photochemistry, and photoinhibition in maize under field conditions. Their results showed that N deficiency resulted in lesser $CO_2$ assimilation capacity and increased susceptibility to photoinhibition. Zhang et al. [30] reported a higher net photosynthetic rate in the flag leaf of wheat due to the N. Cai et al. [31] concluded that the photosynthetic response to the N application rate is variable among wheat cultivars. Similarly, an increase in the total chlorophyll contents of leaves was positively correlated with the N supply [32]. Moran et al. [33] showed that foliar concentration of photosynthetic pigments has a relationship with N supply, and it then ultimately leads to the increased physiological traits of plants. Similar results were depicted by Shrestha et al. [34] in their studies.

Zhong et al. [35] studied the effect of N addition on the sensitivity of photosynthesis among $C_3$ versus $C_4$ grass species under extreme drought and re-watering conditions. Their results concluded that N addition resulted in an increase in biomass, but photosynthesis resilience was lower under

drought conditions. However, faster recovery of photosynthesis was observed due to the addition of N. They further concluded that during drought, the N addition effect on photosynthesis was asymmetric, and it is more specific for the plants which have different photosynthetic nitrogen use efficiency (PNUE). Thus, N could be used to mitigate abiotic stress like drought and could help plants to build resilience to climate change. Similarly, N could help to improve carbon fixation in crops as it is one of the essential components of amino acids. Additionally, it has been well documented that N addition improves the proline, sugars, and antioxidant enzymes in the plants. Sánchez et al. [36] reported an increased proline level under adequate N supply. Similarly, the accumulation of N-containing compounds could lead to the higher survival of plant species under stress [37]. Burns [38], in his findings, showed that N manipulation could help to improve plant growth and development. It has been reported that 90% of the biomass produced by crops is derived from photosynthesis; thus, Makino et al. [39] emphasized the design of strategies that could help to improve crop photosynthesis and grain yield under given N supply. N is the main component of the plant body and part of biological molecules like protein and enzymes, which are involved in different metabolic processes in plants. Thus, deficiency of N could lead to the overall decline in the physiological traits of crops, e.g., net photosynthesis (An), stomatal conductance ($g_s$), biomass, and finally, crop yield. Hence, this study was conducted to explore how different N application rates and methods improve the physiological traits, biomass and grain yield of wheat crop under field conditions. This will be useful for wheat crop management from the perspective of crop physiological processes.

The objectives of the study were (i) to investigate the wheat crop physiological traits (SPAD chlorophyll contents, stomatal conductance ($g_s$), stomatal resistance ($R_s$), transpiration rate (E), net photosynthetic rate ($A_n$)), crop total N, N efficiencies, and agronomic traits in response to N application rates and methods under field conditions and (ii) to see the relationship of physiological characteristics with wheat crop yield only.

## 2. Materials and Methods

### 2.1. Study Sites, Treatments, Plant Material, and Experimental Design

Field experiments were carried out during the wheat growing seasons of 2013–2014 and 2014–2015 at three variable climatic locations of Pothwar, i.e., low rainfall area, Talagang (32°55′ N, 72°25′ E), medium rainfall area, URF-Koont (32°93′ N, 72°86′ E), and high rainfall area, Islamabad (33°40′ N, 73°10′ E) [40] (Figures 1–3). Urea $((NH_2)_2CO)$ (46% N) fertilizer was applied as full at the time of sowing and split doses (sowing, tillering, and anthesis). Treatments include $T_1$ = control (no fertilizer applied), full dose of N applied at the time of crop sowing, i.e., $T_2$ = 50 kg N ha$^{-1}$, $T_3$ = 100 kg N ha$^{-1}$ and $T_4$ = 150 kg N ha$^{-1}$, and split application of N at different timings during different stages of the crop. called split application of N, i.e., $T_5$: application of 50 kg N ha$^{-1}$ (15 kg N ha$^{-1}$ (sowing, BBCH (Biologische Bundesanstalt Bundessortenamt und Chemische Industrie) 0): 20 kg N ha$^{-1}$ (tillering, BBCH 20): 15 Kg N ha$^{-1}$ (anthesis, BBCH 60), $T_6$: application of 100 kg N ha$^{-1}$ (30 kg N ha$^{-1}$ (sowing, BBCH 0): 40 kg N ha$^{-1}$ (tillering, BBCH 20): 30 kg N ha$^{-1}$ (anthesis, BBCH 60) and $T_7$: application of 150 kg N ha$^{-1}$ (45 kg N ha$^{-1}$ (sowing, BBCH 0): 60 kg N ha$^{-1}$ (tillering, BBCH 20): 45 kg N ha$^{-1}$ (anthesis, BBCH 60). Each treatment was replicated three times. The field experiments were laid out using a randomized complete block design (RCBD). The soil physiochemical properties at Islamabad, URF-Koont and Talagang are presented in Tables 1 and 2. The plot size was 5 × 6 m$^2$ in which one bread wheat cultivar (Pakistan-13) was sown on 15th November for two years with a row to row distance of 25 cm. This variety was released by the Pakistan Agriculture Research Council (PARC) with yield potential of 6.0 tones ha$^{-1}$ and good adaptability to the rainfed area. It has 12.1% protein content with gluten contents of 21.5% and 1000 seed weight of 48 g. This cultivar is resistant to yellow rust, leaf rust and stem rust race of Ug99. Weed control was done manually. Sowing was done by hand drill using a seed rate of 50 kg acre$^{-1}$. Prior to sowing, the particular field remained fallow during summer, was ploughed once with a soil inverting implement and thereafter thrice with a tractor mounted

cultivator. A one-meter path was maintained to isolate treatments. Pedigree/parentage of sown cultivar is PTSS02B00132T-0TOPY-0B-0Y-0B-38Y-0M-0SYMEX94.27.1.20/3/SOKOLL//ATTILA/3*BCN, and it was released in the year 2013 for the rainfed conditions of Punjab, Pakistan.

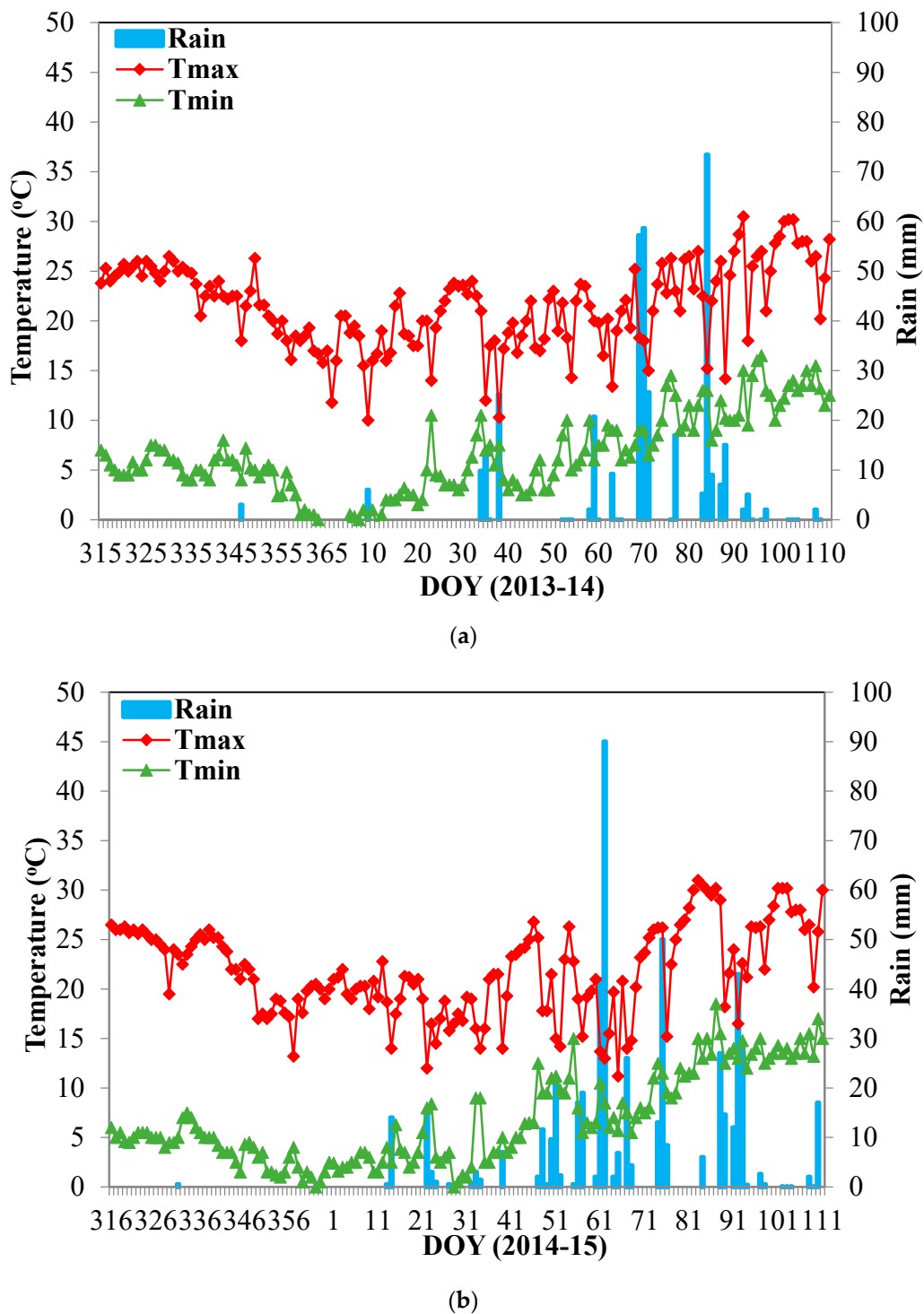

**Figure 1.** Climatic variables (temperature (Tmax and Tmin) and rainfall) during wheat crop growing season for two years (**a**; 2013–14 and **b**; 2014–15) at Islamabad (DOY = days of year).

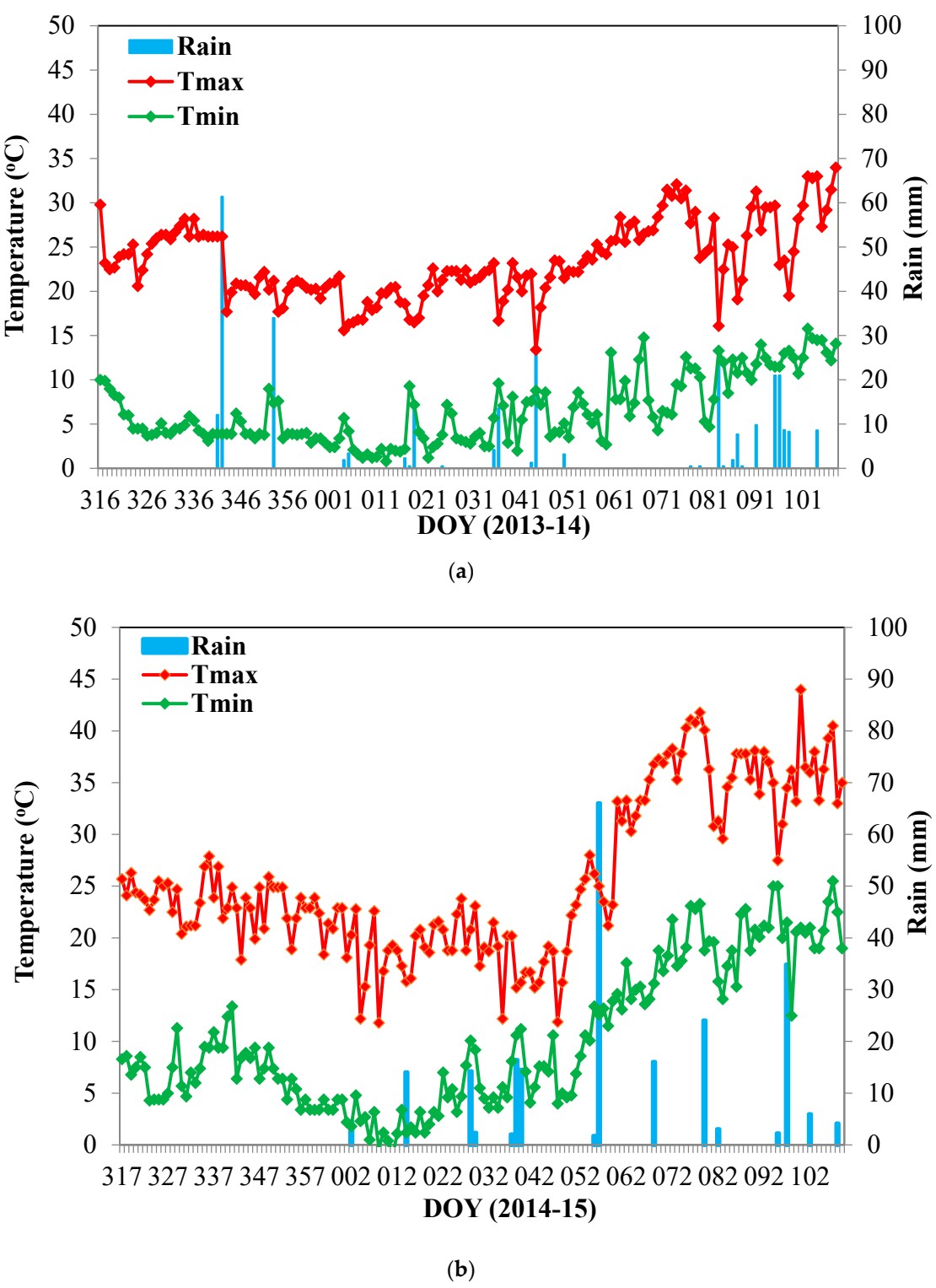

**Figure 2.** Climatic variables (temperature (Tmax and Tmin) and rainfall) during wheat crop growing season for two years (**a**; 2013–14 and **b**; 2014–15) at URF-Koont (DOY = days of year).

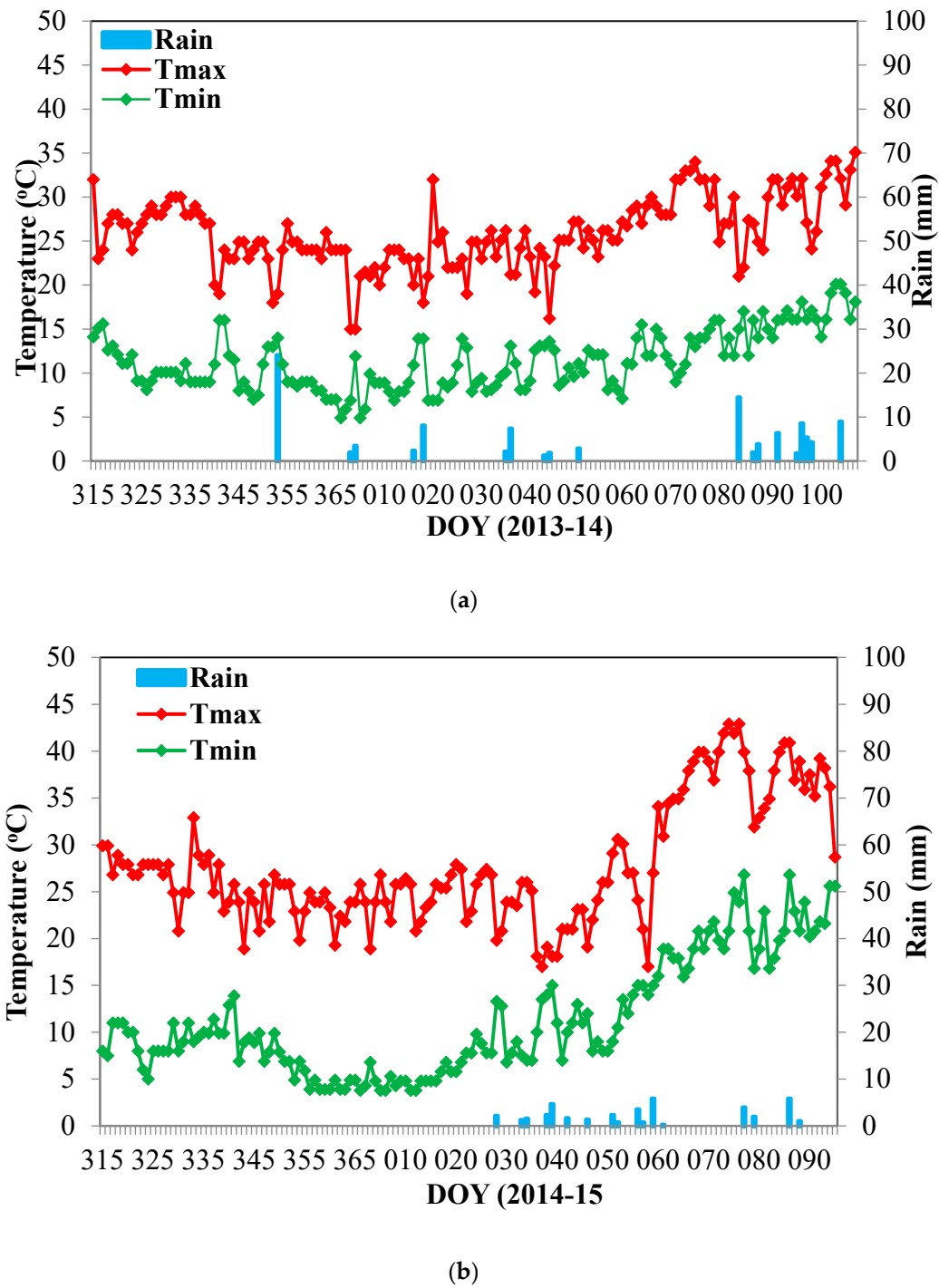

(**a**)

(**b**)

**Figure 3.** Climatic variables (temperature (Tmax and Tmin) and rainfall) during wheat crop growing season for two years (**a**; 2013–14 and **b**; 2014–15) at Talagang (DOY = days of year).

**Table 1.** Physiochemical analysis for three study sites from 0–15 cm, 15–30 cm and 30–45 cm during 2013–14.

| Determinations (2013–14) | Units | Islamabad | | | URF-Koont | | | Talagang | | |
|---|---|---|---|---|---|---|---|---|---|---|
| | | 0–15 | 15–30 | 30–45 | 0–15 | 15–30 | 30–45 | 0–15 | 15–30 | 30–45 |
| pH | 1:01 | 7.5 | 7.6 | 8.3 | 8.1 | 8.6 | 8.8 | 8.3 | 8.4 | 8.5 |
| EC | dSm$^{-1}$ | 0.24 | 0.2 | 0.21 | 0.32 | 0.34 | 0.26 | 0.28 | 0.27 | 0.29 |
| Nitrogen | % | 0.042 | 0.041 | 0.039 | 0.04 | 0.36 | 0.33 | 0.032 | 0.031 | 0.029 |
| Nitrate-N | mg Kg$^{-1}$ | 7.86 | 7.28 | 6.5 | 4.52 | 4.21 | 4.04 | 3.16 | 3.34 | 3.12 |
| AV.P | mg kg$^{-1}$ | 3.68 | 3.46 | 3.41 | 2.8 | 3 | 2.86 | 2 | 2.16 | 2.32 |
| K | mg kg$^{-1}$ | 156 | 175 | 177 | 114 | 150 | 157 | 118 | 108 | 109 |
| Organic Carbon | % | 0.89 | 0.76 | 0.68 | 0.72 | 0.47 | 0.45 | 0.64 | 0.75 | 0.72 |
| Silt | % | 0.35 | 0.35 | 0.35 | 23 | 21 | 20 | 27 | 28 | 28 |
| Sand | % | 0.31 | 0.31 | 0.31 | 56 | 56 | 56 | 58 | 56 | 55 |
| Clay | % | 0.34 | 0.34 | 0.34 | 21 | 23 | 24 | 15 | 16 | 17 |
| Texture | | Loam | Loam | Loam | Sandy clay loam | Sandy clay loam | Sandy clay loam | sandy loam | sandy loam | sandy loam |
| B. Density | gcm$^{-3}$ | 1.24 | 1.42 | 1.46 | 1.31 | 1.58 | 1.66 | 1.51 | 1.59 | 1.68 |
| SLL | mmmm$^{-1}$ | 0.07 | 0.09 | 0.198 | 0.061 | 0.08 | 0.08 | 0.08 | 0.08 | 0.08 |
| SDUL | mmmm$^{-1}$ | 0.38 | 0.30 | 0.30 | 0.28 | 0.23 | 0.22 | 0.18 | 0.18 | 0.15 |
| Saturated SW | mmmm$^{-1}$ | 0.417 | 0.424 | 0.387 | 0.38 | 0.35 | 0.31 | 0.36 | 0.33 | 0.3 |
| SW at tillering (BBCH20) | mmmm$^{-1}$ | 0.30 | 0.24 | 0.23 | 0.24 | 0.18 | 0.18 | 0.15 | 0.15 | 0.14 |
| SW at anthesis (BBCH60) | mmmm$^{-1}$ | 0.33 | 0.25 | 0.23 | 0.24 | 0.19 | 0.19 | 0.16 | 0.16 | 0.15 |

AV.P: available phosphorus, B. Density: bulk density, SLL: soil lower limit, SDUL: soil drain upper limit and SW: soil water.

**Table 2.** Soil physiochemical analysis for three study sites from 0–15 cm, 15–30 cm and 30–45 cm during 2014–15.

| Determinations (2014–15) | Units | Islamabad | | | URF-Koont | | | Talagang | | |
|---|---|---|---|---|---|---|---|---|---|---|
| | | 0–15 | 15–30 | 30–45 | 0–15 | 30–45 | 45–60 | 15–30 | 30–45 | 45–60 |
| pH | 1:01 | 7.4 | 7.5 | 7.9 | 8.2 | 8.1 | 8 | 7.9 | 8 | 7.7 |
| EC | dSm$^{-1}$ | 0.23 | 0.2 | 0.2 | 0.28 | 0.31 | 0.31 | 0.21 | 0.23 | 0.23 |
| Nitrogen | % | 0.041 | 0.042 | 0.039 | 0.03 | 0.02 | 0.02 | 0.031 | 0.036 | 0.035 |
| Nitrate-N | mg Kg$^{-1}$ | 6.4 | 5.9 | 5.3 | 2.58 | 2.77 | 2.64 | 2.61 | 2.51 | 2.4 |
| AV.P | mg kg$^{-1}$ | 3.1 | 2.9 | 3.3 | 2.38 | 2.55 | 2.43 | 1.92 | 2.07 | 2.23 |
| K | mg kg$^{-1}$ | 120 | 135 | 159 | 109 | 143 | 149 | 115 | 101 | 106 |
| Organic Carbon | % | 0.72 | 0.69 | 0.5 | 0.68 | 0.45 | 0.43 | 0.64 | 0.75 | 0.72 |
| Silt | % | 0.35 | 0.35 | 0.35 | 23 | 21 | 20 | 27 | 28 | 28 |
| Sand | % | 0.31 | 0.3 | 0.31 | 56 | 56 | 56 | 58 | 56 | 55 |
| Clay | % | 0.34 | 0.35 | 0.34 | 21 | 23 | 24 | 15 | 16 | 17 |

**Table 2.** *Cont.*

| Determinations (2014–15) | Units | Islamabad | | | URF-Koont | | | Talagang | | |
|---|---|---|---|---|---|---|---|---|---|---|
| | | 0–15 | 15–30 | 30–45 | 0–15 | 30–45 | 45–60 | 15–30 | 30–45 | 45–60 |
| Texture | | Loam | Loam | Loam | Sandy clay loam | Sandy clay loam | Sandy clay loam | sandy loam | sandy loam | sandy loam |
| BD | $gcm^{-3}$ | 1.22 | 1.4 | 1.44 | 1.29 | 1.55 | 1.65 | 1.53 | 1.61 | 1.7 |
| SLL | $mmmm^{-1}$ | 0.09 | 0.198 | 0.186 | 0.057 | 0.076 | 0.076 | 0.07 | 0.08 | 0.08 |
| SDUL | $mmmm^{-1}$ | 0.29 | 0.28 | 0.26 | 0.26 | 0.21 | 0.21 | 0.23 | 0.23 | 0.25 |
| Saturated SW | $mmmm^{-1}$ | 0.424 | 0.419 | 0.359 | 0.38 | 0.33 | 0.29 | 0.36 | 0.33 | 0.3 |
| SW at tillering (BBCH20) | $mmmm^{-1}$ | 0.23 | 0.25 | 0.21 | 0.23 | 0.17 | 0.17 | 0.20 | 0.20 | 0.22 |
| SW at anthesis (BBCH60) | $mmmm^{-1}$ | 0.26 | 0.26 | 0.22 | 0.24 | 0.18 | 0.18 | 0.21 | 0.21 | 0.23 |

AV.P: available phosphorus, B. Density: bulk density, SLL: soil lower limit, SDUL: soil drain upper limit and SW: soil water.

## 2.2. Physiological Traits Measurements

Wheat physiological traits, which include $g_s$ (stomatal conductance), $R_s$ (stomatal resistance) ($m^2$ s $mole^{-1}$), net photosynthetic rate ($A_n$) (μ mole/$m^2$/s), E (transpiration rate) (mmole $H_2O$ $m^{-2}$ $s^{-1}$) and Ci (intercellular $CO_2$) (μmole $CO_2$/mol air) were measured with an infrared gas analyzer (IRGA) at anthesis (BBCH 60) stage of wheat by putting a leaf into the chamber and adjusting its leaf area, and when the value became constant, it was recorded. The averages of five samples were taken as measured by Long and Bernacchi [41]. Similarly, chlorophyll contents were measured using a SPAD chlorophyll meter. The chlorophyll contents were taken from the top, middle, and base of leaves, and then the average value was used to represent SPAD chlorophyll contents. Proline content (μg $g^{-1}$) was measured by taking fresh leaves (0.5 g) from plants at the flag leaf stage from each plot. The samples were normalized in ten ml of three percent sulfosalicyclic acid ($C_7H_6O_6S$) and then filtered. The stress amino acid, proline ($C_5H_9NO$), was estimated spectrophotometrically following the ninhydrin method [42].

## 2.3. Crop Total Nitrogen

The amount of N in the plant was determined at tillering (BBCH 20), anthesis (BBCH 60), and maturity stages (BBCH 90). The one-meter square area was used to take plant samples. Nitrogen contents from plant samples were determined after oven drying at 65 °C for 48 h. After drying, samples were ground using a Wiley Mill, and samples were placed in plastic bottles to determine N contents. Samples of 2 g in 30 to 50 mL of acid and approximately 100 mL sodium hydroxide (NaOH) solutions were used. The TN (Total nitrogen) was measured after wet digestion in concentrated sulphuric acid ($H_2SO_4$) using the Kjeldahl procedure [43]. The total organic nitrogen was converted to ammonium sulfate. Ammonia formed was distilled into boric acid solution under alkaline conditions. The borate anions formed were titrated with standardized hydrochloric acid. This was used to calculate the content of nitrogen, representing the amount of crude protein in the sample, which is generally 16% of nitrogen, thus the conversion factor used was 6.25.

*2.4. Nitrogen Efficiencies*

Rahimizadeh et al.'s [44] procedure was used to determine the nitrogen uptake efficiency (NUpE), nitrogen utilization efficiency (NUtE), and nitrogen use efficiency (NUE):

$$NUpE = \frac{N_T}{N_{supply}}$$

$$NUtE = \frac{G_Y}{N_T}$$

$$NUE = \frac{G_Y}{N_{supply}}$$

where $N_T$ = total plant $N$ uptake, $G_Y$ = grain yield, and $S_{upply}$ = sum of soil $N$ content at sowing and $N$ fertilizer.

*2.5. Agronomic Traits*

At physiological maturity, total numbers of fertile tillers were counted from an area of one $m^2$ from each plot. Furthermore, a sub-sample of a thousand grains from each treatment was weighed using a digital weighing balance. The biological yield was measured by harvesting a one $m^2$ area per plot and was converted to get the final yield in kg ha$^{-1}$. Grain yield was calculated by harvesting a one $m^2$ area per plot and was converted to get final yield in kg ha$^{-1}$. Finally, the harvest index (HI) was measured using the following formula.

$$Harvest\ index\ (HI) = \frac{Grain\ yield}{Biological\ Yield} \times 100$$

*2.6. Statistical Analysis*

The significance of the effects of years, locations, and N treatments at the 0.05 level was determined using a three-way analysis of variance (ANOVA) using SPSS 19.0 software (SPSS, Inc., Chicago, IL, USA). Means obtained were compared using a Tukey test at 0.05% level of significance.

## 3. Results

*3.1. Crop Physiological Traits*

A significant difference was found for all physiological characteristics against all treatments. Stomatal conductance ($g_s$) results showed that it remained significantly higher during 2013–14 as compared to 2014–15. Among the sites, the maximum $g_s$ was observed for Islamabad, followed by URF-Koont, while it was minimum at Talagang, i.e., a low rainfall and high-temperature location. N treatment effects on stomatal conductance revealed that it was maximum at $T_4$, while the minimum was observed for the control treatment with full application of N at sowing. However, among split N application, the highest $g_s$ was recorded for $T_6$, which was at par with $T_7$ (Table 3). The interactive effect of Y × L was significant on $g_s$, while all other interactions were non-significant. The results for stomatal resistance ($R_s$) were inverse to $g_s$. Among the years, the highest $R_s$ (1.05 $m^2$ s mole$^{-1}$) was observed during 2014–15, while at sites, it remained high at Talagang, followed by URF-Koont and Islamabad. N treatment's impacts on $R_s$ revealed that with the application of higher N, it decreases significantly. The lowest stomatal resistance was found for $T_5$ and $T_6$, while the highest stomatal resistance was observed for control treatment $T_1$. Only Y × L interactions were found to be significant for stomatal resistance, while all other interactions were non-significant. Net Photosynthesis ($A_n$) results showed that it remained highest for the first year compared to 2014–15. Among the sites, the maximum $A_n$ was recorded for Islamabad, while it remained the minimum at Talagang. N treatments have shown significant impacts on $A_n$, and the highest $A_n$ was observed for split treatment $T_6$, which

was at par with $T_7$. However, among the full dose application of N maximum, $A_n$ was found for $T_4$, while it was minimum for $T_1$. Transpiration rate (E) remained highest during 2013–14, while the lowest E was recorded for 2014–15. Among the sites, the highest E was recorded for Islamabad, followed by URF-Koont and Talagang. The impact of N treatments on the transpiration rate revealed that it remained highest for split application of N, i.e., $T_6$, which was at par with $T_7$ as well as with $T_4$. However, the lowest E was observed for the control treatment ($T_1$). A similar trend was observed for Ci ($\mu$mole $CO_2$ $mol^{-1}$ air) and SPAD chlorophyll contents. However, proline contents remained maximum for the second year compared to the first year. Among the sites, the highest proline contents were observed at Talagang, while it remained lowest at Islamabad. N addition resulted in the positive effects on proline contents, and the highest proline was observed for split treatment $T_6$.

**Table 3.** Physiological traits of wheat for two years, at three variable sites and under different nitrogen treatments; *** $p \leq 0.001$; NS = non-significant.

| | gs (mole $H_2O$ $m^{-2}$ $s^{-1}$) | Rs ($m^2$ s $mole^{-1}$) | An $\mu$mole $CO_2$ $m^{-2}$ $s^{-1}$ | E (mmole $H_2O$ $m^{-2}$ $s^{-1}$) | Ci ($\mu$mole $CO_2$ $mol^{-1}$ air) | Chlorophyll Contents (SPAD) | Proline Content ($\mu$g $g^{-1}$) |
|---|---|---|---|---|---|---|---|
| **Years (Y)** | | | | | | | |
| **2013–14** | 0.65 [a] | 0.73 [b] | 18.70 [a] | 7.52 [a] | 298.41 [a] | 54.672 [a] | 30.77 [b] |
| **2014–15** | 0.50 [b] | 1.05 [a] | 16.24 [b] | 6.32 [b] | 272.60 [b] | 45.56 [b] | 38.25 [a] |
| **Study Sites/Locations (L)** | | | | | | | |
| **Islamabad** | 0.67 [a] | 0.66 [c] | 18.88 [a] | 7.95 [a] | 317.79 [a] | 51.80 [a] | 27.85 [c] |
| **URF-Koont** | 0.56 [b] | 0.87 [b] | 16.77 [b] | 6.89 [b] | 290.12 [b] | 46.79 [b] | 33.86 [b] |
| **Talagang** | 0.46 [c] | 1.07 [a] | 15.55 [c] | 6.13 [c] | 252.60 [c] | 40.29 [c] | 40.13 [a] |
| **Nitrogen Treatments (T)** | | | | | | | |
| **$T_1$ = $N_0$** | 0.60 [c] | 0.71 [a] | 15.25 [c] | 6.82 [c] | 290.79 [d] | 44.51 [d] | 27.97 [d] |
| **$T_2$ = $N_{50}$** | 0.73 [b] | 0.56 [b] | 17.50 [b] | 8.15 [b] | 310.18 [c] | 48.39 [c] | 29.52 [c] |
| **$T_3$ = $N_{100}$** | 0.78 [a] | 0.49 [c] | 18.10 [b] | 8.75 [b] | 318.33 [b] | 54.55 [b] | 32.15 [b] |
| **$T_4$ = $N_{150}$** | 0.79 [a] | 0.48 [c] | 19.49 [a] | 9.22 [a] | 320.23 [b] | 57.28 [a] | 33.82 [b] |
| **$T_5$ = Split $N_{50}$** | 0.73 [b] | 0.56 [b] | 17.52 [b] | 8.15 [b] | 310.19 [c] | 54.39 [b] | 30.74 [c] |
| **$T_6$ = Split $N_{100}$** | 0.80 [a] | 0.48 [c] | 20.07 [a] | 9.58 [a] | 329.25 [a] | 58.86 [a] | 35.42 [a] |
| **$T_7$ = Split $N_{150}$** | 0.80 [a] | 0.48 [c] | 19.79 [a] | 9.31 [a] | 325.27 [a] | 58.88 [a] | 35.42 [a] |
| **Interactions** | | | | | | | |
| **Y × L** | *** | *** | *** | *** | NS | NS | *** |
| **Y × T** | NS | NS | NS | NS | NS | NS | NS |
| **L × T** | NS | NS | NS | NS | NS | *** | NS |
| **Y × L × T** | NS | NS | NS | NS | NS | NS | NS |

gs: stomatal conductance, $R_s$: stomatal resistance, $A_n$: net photosynthetic rate, E: transpiration rate and Ci: intercellular $CO_2$. Different letters (a,b,c) in each column indicate significant difference of averages among each other while similar shows non-significant difference among each other.

## 3.2. Crop Total Nitrogen

Significant variation for crop/biomass total nitrogen (TN) was observed during both years (2013–14 and 2014–15) at three locations under different N treatments. Both years differed significantly for crop N at the tillering stage. Maximum TN (2.81 kg $ha^{-1}$) was observed during 2013–14, while minimum TN (2.68 kg $ha^{-1}$) was recorded during 2014–15 (Table 4). During 2013–14, 5% higher TN was recorded than 2014–15. At tillering among locations, maximum TN (3.61 kg $ha^{-1}$) was recorded at Islamabad, while minimum TN (1.99 kg $ha^{-1}$) was observed at Talagang. There was a 44% variation among the study sites for TN at the tillering stage. The maximum TN (4.16 kg $ha^{-1}$) was recorded under $T_4$, while minimum TN at tillering was recorded under $T_5$ (1.84 kg $ha^{-1}$). Under $T_4$, 42% higher TN was registered than $T_5$ at the tillering stage. The interactive effects of years × locations (Y × L) and locations × treatments (L × T) were highly significant, while Y × T and Y × L × T remained non-significant. For interactive effect, Y × L maximum crop TN was accumulated during 2014–15 (3.77 kg $ha^{-1}$) at Islamabad, while minimum crop TN (1.82 kg $ha^{-1}$) was recorded during 2014–15 at Talagang (Figure 4). Maximum TN was recorded at Islamabad under $T_4$ (5.46 kg $ha^{-1}$), while minimum TN at tillering was recorded (1.34 kg $ha^{-1}$) at Talagang under $T_5$ (Figure 4). Similarly, a significant difference was noted during both years at the anthesis stage for TN. Maximum TN (33.53 kg $ha^{-1}$) was recorded during 2013–14, while minimum crop TN (31.51 kg $ha^{-1}$) was observed during 2014–15. During 2013–14, 6% higher N was noted than 2014–15. A significant difference in TN was recorded at three diverse

locations. Higher TN (39.93 kg ha$^{-1}$) was observed at Islamabad, while lower TN (23.39 kg ha$^{-1}$) was recorded at Talagang. The difference in TN among Islamabad and Talagang was 41%. Total N also differed under different N treatments. Maximum crop TN (44.23 kg ha$^{-1}$) was recorded under $T_7$, while minimum TN (17.64 kg ha$^{-1}$) was observed under $T_1$. There was 88% higher TN for $T_7$ compared to $T_1$. The interactive effects $Y \times L$ and $L \times T$ were highly significant while $Y \times T$ and $Y \times L \times T$ remained non-significant. For interactive effect, $L \times T$ maximum crop TN was accumulated during 2013–14 (42.71 kg ha$^{-1}$) at Islamabad, while minimum crop TN was recorded (22.91 kg ha$^{-1}$) during 2013–14 at Talagang (Figure 4). For interactive effect, $Y \times L$ highest crop TN (55.59 kg ha$^{-1}$) was recorded during 2013–14 under $T_7$, while the lowest TN (12.39 kg ha$^{-1}$) was recorded at Talagang under control N treatment (Figure 4).

**Table 4.** Crop total N (kg N ha$^{-1}$) at tillering (BBCH20), anthesis (BBCH60) and maturity (BBCH90) stages of wheat crop during two years at three study sites under seven nitrogen treatments with significance of their interactions; *** $p \leq 0.001$; NS = non-significant.

| | Crop N at Tillering (BBCH20) (kg N ha$^{-1}$) | Crop N at Anthesis (BBCH60) (kg N ha$^{-1}$) | Crop N at Maturity (BBCH90) (kg N ha$^{-1}$) |
|---|---|---|---|
| **Years (Y)** | | | |
| **2013–14** | 2.81 [a] | 33.53 [a] | 66.417 [a] |
| **2014–15** | 2.68 [b] | 31.51 [b] | 62.95 [b] |
| **Study Sites/Locations (L)** | | | |
| **Islamabad** | 3.61 [a] | 39.93 [a] | 87.22 [a] |
| **URF-Koont** | 2.64 [b] | 34.25 [b] | 63.60 [b] |
| **Talagang** | 1.99 [c] | 23.38 [c] | 43.22 [c] |
| **Nitrogen Treatments (T)** | | | |
| **$T_1 = N_0$** | 2.11 [e] | 17.64 [f] | 32.91 [e] |
| **$T_2 = N_{50}$** | 2.87 [c] | 25.64 [e] | 48.60 [d] |
| **$T_3 = N_{100}$** | 3.54 [b] | 31.61 [d] | 64.17 [c] |
| **$T_4 = N_{150}$** | 4.16 [a] | 36.62 [b] | 75.32 [b] |
| **$T_5 = $ Split $N_{50}$** | 1.84 [f] | 33.70 [c] | 63.81 [c] |
| **$T_6 = $ Split $N_{100}$** | 2.17 [e] | 38.18 [b] | 78.74 [b] |
| **$T_7 = $ Split $N_{150}$** | 2.52 [d] | 44.23 [a] | 89.20 [a] |
| **Interactions** | | | |
| **$Y \times L$** | *** | *** | *** |
| **$Y \times T$** | NS | NS | *** |
| **$L \times T$** | *** | *** | *** |
| **$Y \times L \times T$** | NS | NS | *** |

Different alphabets (a,b,c,d,e and f) in each column depicts significant difference of averages among each other.

A significant variation for crop TN was observed during both years (2013–14 and 2014–15) at three varying climatic locations under different N treatments at maturity. Both years differed significantly for crop TN at the maturity stage. Maximum TN (66.42 kg ha$^{-1}$) was observed during 2013–14, while minimum TN (62.95 kg ha$^{-1}$) was recorded during 2014–15 (Table 4). During 2013–14, 5% higher TN was recorded than 2014–15. At maturity among locations, maximum TN (87.22 kg ha$^{-1}$) was recorded at Islamabad, while minimum crop TN (43.22 kg ha$^{-1}$) was observed at Talagang. There was a 47% variation among study sites for TN at the maturity stage. Meanwhile, maximum TN (89.20 kg ha$^{-1}$) was recorded under $T_7$, while minimum TN at maturity was recorded under control N treatment (32.91 kg ha$^{-1}$). Under N treatment $T_7$, 46% higher TN was registered than $T_1$ at the maturity stage. The interactive effects $Y \times L$ and $L \times T$ were highly significant, while $Y \times T$ and $Y \times L \times T$ remained non-significant. For interactive effect, $Y \times L$ maximum crop TN was accumulated during 2014–15 (89.63 kg ha$^{-1}$) at Islamabad followed by 2013–14 at Islamabad (84.81 kg ha$^{-1}$), while minimum crop

TN (37.53 kg ha$^{-1}$) was recorded during 2014–15 at Talagang (Figure 4). Maximum TN was recorded at Islamabad under T$_7$ (123.19 kg ha$^{-1}$), while minimum TN at maturity was recorded (25.55 kg ha$^{-1}$) at Talagang under T$_1$ (Figure 4).

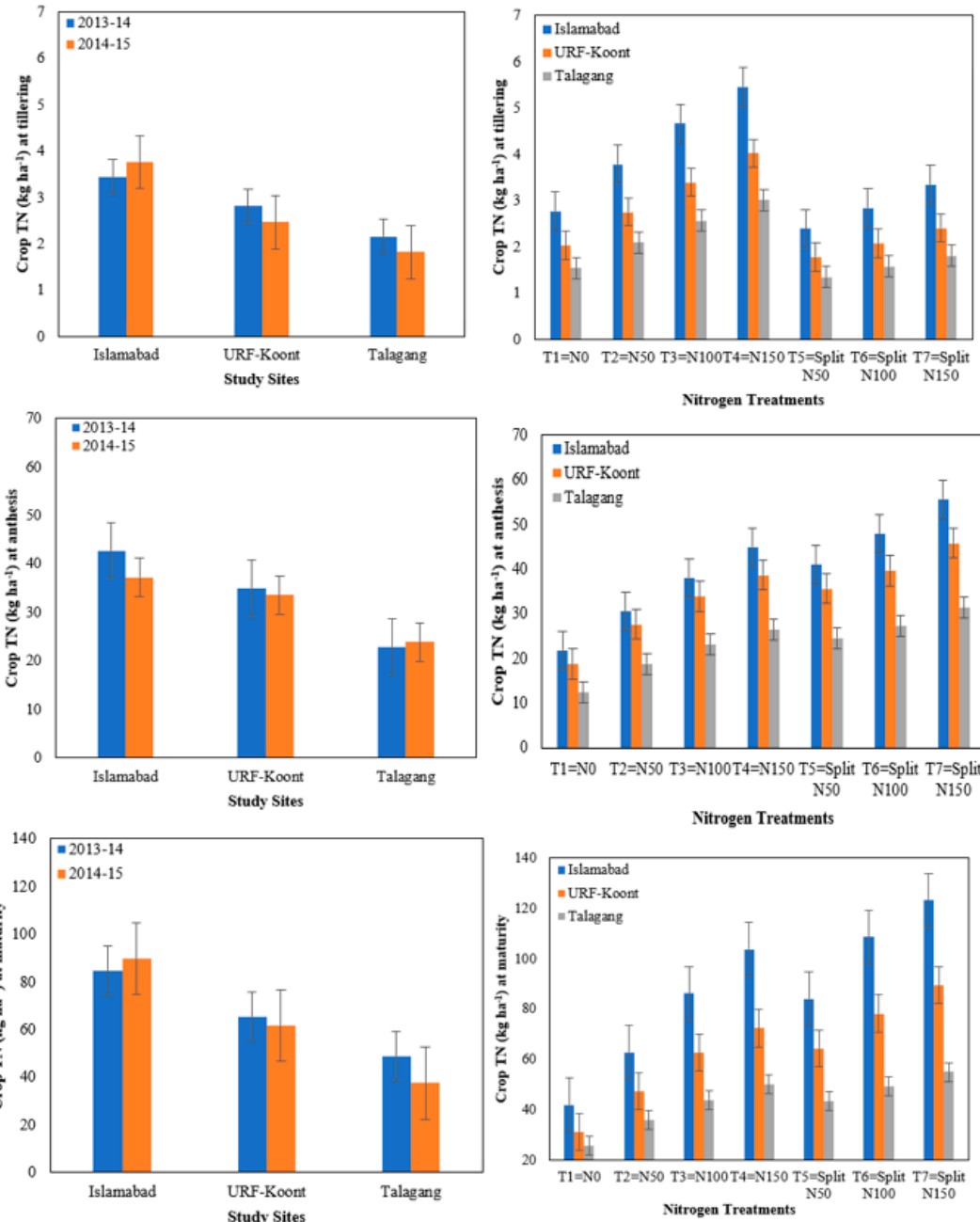

**Figure 4.** Crop total nitrogen (TN) Kg N ha$^{-1}$ at tillering, anthesis and at maturity stages for years × locations (Y × L) and locations × treatments (L × T) interactions.

*3.3. Nitrogen Efficiencies*

3.3.1. Nitrogen Use Efficiency (kg kg$^{-1}$)

A significant variation for NUE was observed during both years (2013–14 and 2014–15) at three varying climatic locations under different N treatments. Both years differed significantly for NUE. Maximum NUE (18.35 kg kg$^{-1}$) was observed during 2013–14, while minimum NUE (16.63 kg kg$^{-1}$) was recorded during 2014–15 (Table 5). During 2013–14, 9% higher NUE was recorded than 2014–15. Among locations, maximum NUE (22.96 kg kg$^{-1}$) was recorded at Islamabad, while minimum NUE (11.31 kg kg$^{-1}$) was observed at Talagang. There was a 51% variation among study sites for NUE. The maximum NUE (21.72 kg kg$^{-1}$) was recorded under $T_1$, while the minimum NUE was recorded under $T_7$ N treatment (13.395 kg kg$^{-1}$). Under N treatment $T_1$, 38% higher NUE was registered than $T_7$. The interactive effects viz. Y × L, Y × T, L × T, and Y × L × T remained statistically non-significant for NUE.

**Table 5.** Wheat nitrogen use efficiencies (nitrogen use (kg kg$^{-1}$), utilization and uptake efficiencies) crop during two years at three study sites under seven nitrogen treatments with significance of their interactions; *** $p \leq 0.001$; NS = non-significant.

| Years (Y) | NUE | NUtE | NUpE |
|---|---|---|---|
| **2013–14** | 18.35 [a] | 44.18 [b] | 0.41 [a] |
| **2014–15** | 16.63 [b] | 44.35 [a] | 0.39 [b] |
| **Study Sites/Locations (L)** | | | |
| **ISLAMABAD** | 22.96 [a] | 43.53 [b] | 0.53 [a] |
| **URF-Koont** | 18.22 [b] | 47.08 [a] | 0.39 [b] |
| **Talagang** | 11.31 [c] | 42.20 [c] | 0.27 [c] |
| **Nitrogen Treatments (T)** | | | |
| **$T_1 = N_0$** | 21.72 [a] | 52.78 [b] | 0.41 [bc] |
| **$T_2 = N_{50}$** | 20.03 [b] | 53.98 [a] | 0.37 [d] |
| **$T_3 = N_{100}$** | 16.49 [c] | 46.33 [c] | 0.35 [de] |
| **$T_4 = N_{150}$** | 13.48 [d] | 41.38 [d] | 0.32 [e] |
| **$T_5 =$ Split $N_{50}$** | 20.03 [b] | 40.84 [e] | 0.49 [a] |
| **$T_6 =$ Split $N_{100}$** | 17.31 [c] | 39.42 [f] | 0.43 [b] |
| **$T_7 =$ Split $N_{150}$** | 13.39 [d] | 35.15 [g] | 0.38 [cd] |
| **Interactions** | | | |
| **Y × L** | NS | *** | *** |
| **Y × T** | NS | NS | *** |
| **L × T** | NS | *** | *** |
| **Y × L × T** | NS | NS | *** |

NUE: nitrogen use efficiency, NUtE: nitrogen utilization efficiency and NUpE: nitrogen uptake efficiency: Different letters (a,b,c,d,e, f and g) in each column indicate significant difference of averages among each other while similar shows non-significant difference among each other.

3.3.2. Nitrogen Uptake Efficiency (NUpE)

A significant variation for NUpE was observed during both years (2013–14 and 2014–15) at three diverse locations under different N treatments. Both years differed significantly for NUpE. Maximum NUpE (0.41) was observed during 2013–14, while minimum NUpE (0.39) was recorded during 2014–15 (Table 5). During 2013–14, 5% higher NUpE was recorded than 2014–15. Among locations, maximum NUpE (0.53) was recorded at Islamabad, while minimum NUpE (0.27) was observed at Talagang. There was a 49% variation among study sites for NUpE. Meanwhile, maximum

NUpE (0.49) was recorded under $T_5$, while minimum NUpE was recorded under $T_4$ N treatment (0.32). Under N treatment $T_5$, 35% higher NUpE was recorded than $T_4$.

### 3.3.3. Nitrogen Utilization Efficiency (NUtE) (kg kg$^{-1}$)

A significant variation for (NUtE) was observed during both years (2013–14 and 2014–15) at three varying climatic locations under different N treatments. Both years differed significantly for N utilization efficiency at the maturity stage. Maximum NUtE (44.36 kg kg$^{-1}$) observed during 2013–14 while minimum NUtE (44.19 kg kg$^{-1}$) recorded during 2014–15 (Table 5). During 2013–14, 0.3% higher NUtE was recorded than 2014–15. At maturity, among locations, maximum N utilization efficiency (47.09 kg kg$^{-1}$) was recorded at URF-Koont, while minimum N utilization efficiency (42.2 kg kg$^{-1}$) was observed at Talagang. There was a 10% variation among study sites for N utilization efficiency at the maturity stage. Similarly, maximum NUtE (53.99 kg kg$^{-1}$) was recorded under $T_2$, while minimum NUtE at maturity was recorded under $T_7$ (35.16 kg kg$^{-1}$). Under N treatment $T_2$, 34% higher N utilization efficiency was recorded than $T_7$ at the maturity stage. The interactive effects $Y \times L$, $L \times T$, $Y \times T$, and $Y \times L \times T$ for NUtE were highly significant (Table 5).

### 3.4. Agronomic Traits

Results showed that the number of tillers remained non-significant during both years (2013–14 and 2014–15), while a significant difference was observed at the three different sites under different N treatments (Table 6). Similarly, thousand-grain weight (TGW) remained non-significant during both study years. However, it was significantly different at sites and under different N treatments. Under N treatment $T_6$, 32% higher TGW was recorded than $T_1$. The interactive effects $Y \times L$ was significant while $L \times T$, $Y \times T$, and $Y \times L \times T$ were non-significant. Among locations, maximum TGW (34.3 g) was recorded at Islamabad, while minimum TGW (24.7 g) was observed at Talagang. There was a 28% variation among study sites for TGW. Meanwhile, maximum TGW (34.5 g) was recorded under the treatment $T_6$, while the minimum TGW was recorded under control treatment. Significant variation for the grain yield was observed during both years (2013–14 and 2014–15) at three varying locations under different N treatments. The highest biological yield (9380.9 kg ha$^{-1}$) was recorded for the first year (2013–14) while it was lowest (8704.7 kg ha$^{-1}$) for the second year (2014–15). Among locations, biological yield remained maximum at Islamabad, while the addition of N resulted in the maximum biomass under split application of N, i.e., $T_6$. Maximum grain yield (3001.9 kg ha$^{-1}$) was observed during 2013–14, while minimum grain yield (2611.40 kg ha$^{-1}$) was observed during 2014–15 (Table 6). During 2013–14, a 10% higher grain yield was recorded than 2014–15. Among locations, the highest grain yield (3957.5 kg ha$^{-1}$) was recorded at Islamabad, while the lowest grain yield (1760.6 kg ha$^{-1}$) was detected at Talagang. There was a 52% variation among the study sites for the grain yield. Meanwhile, the highest grain yield (3517.2 kg ha$^{-1}$) was recorded under $T_6$, while the lowest grain yield was recorded under $T_1$ N treatment (1737.8 kg ha$^{-1}$). For N treatment $T_6$, a 44% higher grain yield was recorded than $T_1$. The interactive effects of $L \times T$ were highly significant, while $Y \times L$, $Y \times T$, and $Y \times L \times T$ were non-significant. A considerable difference for the harvest index was observed during both years (2013–14 and 2014–15). During 2013–14, a 6% higher harvest index was recorded than 2014–15. Among locations, the maximum harvest index (0.35) was recorded at Islamabad, while the minimum harvest index (0.25) was observed at Talagang. There was a 29% variation among study sites for the harvest index. Meanwhile, the maximum harvest index (0.32) was recorded under $T_2$, $T_3$, $T_4$, and $T_7$, while the minimum harvest index was recorded under $T_1$ and $T_5$ N treatment (0.29).

**Table 6.** Agronomic traits of wheat crop during two years at three study sites under seven nitrogen treatments with significance of their interactions; *** $p \leq 0.001$; NS = non-significant.

| | No of Tillers m$^{-2}$ | 1000 Grain Weight (gm) | Biological Yield (kg ha$^{-1}$) | Grain Yield (kg ha$^{-1}$) | Harvest Index |
|---|---|---|---|---|---|
| **Years (Y)** | | | | | |
| **2013–14** | 222.4 [NS] | 31.3 [NS] | 9380.9 [a] | 3001.9 [a] | 0.32 [a] |
| **2014–15** | 212.7 | 30.7 | 8704.7 [b] | 2611.4 [b] | 0.3 [b] |
| **Study Sites/Locations (L)** | | | | | |
| **Islamabad** | 236.5 [a] | 34.3 [a] | 10,450.1 [a] | 3957.5 [a] | 0.35 [a] |
| **URF-Koont** | 226.9 [a] | 33.9 [a] | 8578.2 [b] | 2830.8 [b] | 0.33 [b] |
| **Talagang** | 189.2 [b] | 24.7 [b] | 7042.4 [c] | 1760.6 [c] | 0.25 [c] |
| **Nitrogen Treatments (T)** | | | | | |
| **T$_1$ = N$_0$** | 184.5 [c] | 23.5 [c] | 5992.4 [c] | 1737.8 [c] | 0.29 [c] |
| **T$_2$ = N$_{50}$** | 205.8 [b] | 28.1 [b] | 8140.6 [b] | 2605 [b] | 0.32 [a] |
| **T$_3$ = N$_{100}$** | 218.8 [ab] | 31.7 [a] | 9275.9 [a] | 2968.3 [a] | 0.32 [a] |
| **T$_4$ = N$_{150}$** | 229.3 [a] | 33.1 [a] | 10,942.8 [a] | 3501.7 [a] | 0.32 [a] |
| **T$_5$ = Split N$_{50}$** | 221.1 [ab] | 31.6 [a] | 8982.7 [b] | 2605 [b] | 0.29 [c] |
| **T$_6$ = Split N$_{100}$** | 234.4 [a] | 34.5 [a] | 11,345.5 [a] | 3517.2 [a] | 0.31 [b] |
| **T$_7$ = Split N$_{150}$** | 229.1 [a] | 34.3 [a] | 10,976.3 [a] | 3512.4 [a] | 0.32 [a] |
| **Interactions** | | | | | |
| **Y × L** | *** | NS | *** | NS | *** |
| **Y × T** | NS | NS | NS | NS | *** |
| **L × T** | NS | *** | NS | *** | *** |
| **Y × L × T** | NS | NS | NS | NS | *** |

Different letters in each column indicate significant difference of averages among each other while similar shows non-significant difference among each other.

### 3.5. Relationship of Physiological Traits with Grain Yield

Linear regression analysis was performed to see the relationship between grain yield and physiological characteristics combined over the years, locations, and N treatments. The results showed that physiological traits (e.g., $g_s$, $R_s$, $A_n$, E, and SPAD chlorophyll contents) have a significant relationship with grain yield of wheat. The regression equation for grain yield with stomatal conductance showed a positive trend with $R^2$ = 0.98. The equation obtained for grain yield and $g_s$ is presented below.

$$Grain\ yield = -1476.28 + 6518.68 g_s$$

The inverse relationship between grain yield and stomatal resistance was observed with $R^2$ = 0.98. The equation obtained was

$$Grain\ yield = 5353.31 - 3591.18 R_s.$$

A positive, strong association was obtained for net photosynthesis and grain yield ($R^2$ = 0.99). The regression equation for this relationship is

$$Grain\ yield = -5614.79 + 433.27 A_n.$$

Transpiration rate outcomes revealed that with the increase in transpiration rate, grain yield of wheat crop increases significantly ($R^2$ = 0.98). The equation obtained for this trend is

$$Grain\ yield = -3490.54 + 833.24 E.$$

A similar pattern was observed for intercellular carbon dioxide concentration and SPAD chlorophyll contents with $R^2$ values of 0.98 and 0.97, respectively. The model equations were

$$Grain\ yield = -6828.66 + 32.06 C_i$$

$$Grain\ yield = -1501.84 + 74.45\ SPAD_{chlorphyll\ contents}.$$

## 4. Discussion

Climate extremes in the form of rainfall variability, drought, and rise in temperature are the primary abiotic stresses affecting wheat physiological traits, crop total N, N efficiencies, and agronomic characteristics. Our results showed that the physiology of wheat crop decreases under stress (water, temperature, or nutrient); however, this stress could be managed by the application of N [8]. Water stress resulted in a significant decrease in the net photosynthetic rate, crop biomass and growth rate but the decrease was less in N primed treatment. This was because N treatment resulted in enhanced root growth and higher leaf water content. Stomatal conductance ($g_s$) could be the critical determinant of crop yield and productivity as it balances $CO_2$ uptake and water loss. It impacts on the total rate of photosynthesis and water use during the crop growing period. Ahmed et al. [45] concluded that crop physiological traits have a strong association with prevailing climatic conditions. Higher temperatures and lower availability of water resulted in a decline in physiological characteristics like $g_s$, $A_n$, and E but an increase in stomatal resistance. However, they suggested that a change in the sowing date could be an option to mitigate the effect of climatic variables on crop physiology, while our findings confirm that the addition of N as a split application could build resilience in crop physiological traits. Split application at critical growth stages of the crop improves uptake efficiency and minimizes N loss. Since N helps in the process of photosynthesis, its addition could thus be beneficial for the crop. Higher photosynthetic rates lead to higher biomass production and grain yield. Yu et al. [46] reported that photosynthesis and transpiration are interdependent: the improvements of one are linked with the development of others. Higher values for both photosynthesis and transpiration rate under split application of N demonstrates that N helps to maintain water contents in the leaf, thus resulting iin the optimization of $A_n$, E, $g_s$, and $C_i$. Kimball et al. [47] reported reduced stomatal conductance (33–50%) and transpiration rate (20–27%) with the change in the microclimate of the crop. Our results of N addition and its relationship with photosynthetic capacity was at par with the earlier findings in which they concluded that N addition could help to replenish the negative effect of stress by building resilience in the physiological traits of the crop [14,21–23]. Like the present work, an increased net photosynthetic rate ($A_n$) was observed due to the application of N [34]. Gyuga et al. [48] reported that N greatly influences photosynthetic processes, and deficiency of N leads to the declined photosynthesis. Similarly, Abid et al. [49] in their findings concluded that management of N nutrition could build drought tolerance in wheat by maintaining higher photosynthetic activities and antioxidative defense system during vegetative growth periods. A primary driving force for dry matter production in photosynthesis; thus, its management through the optimization of N could help to improve dry matter production in the plant. A positive correlation between leaves photosynthetic capacity and N contents have been shown in earlier work [14,21–23]. Application of N in wheat plants resulted in the improved properties of photosynthetic pigments and increased net photosynthetic rate ($A_n$) [24,25]. Similarly, the fitness of plants could be determined by indicators like chlorophyll content and photosynthetic rate. Thus, moderate stress (drought or nutrients) could decrease photosynthesis, mainly due to stomatal limitations [50–52]. Hence, we suggest here that the application of N in split dosage be used tackle this limitation. Variability in SPAD chlorophyll content due to temperature and water stress were reported in earlier work, and the use of suitable genotypes and optimum sowing time to provide suitable environmental conditions for the crop was suggested [53]. $CO_2$ is an important ecological factor for plant matter, being directly involved in photosynthesis; however, its optimum fixation is linked with stomatal traits and chlorophyll contents in the leaves [54]. Proline is a vital stress defender, and in

our findings, it has been revealed that N addition results in the increased N compound, i.e., proline (Table 3). Improved tolerance in plants was observed due to the accumulation of proline under water and temperature stress [55–59], which resulted in improved crop physiological processes. The role of antioxidants like proline in plant drought tolerance was well-reviewed by Laxa et al. [60]. The higher survival rate of plants under stress has a strong link with the accumulation of N-containing compounds, and their concentration is increased due to the addition of N, as reported in our studies [36,37].

Lu et al. [29] reported that reduced N supply could lead to severe plant growth and lower grain yield. The addition of N could help to increase total N in a plant, as elaborated in our findings (Table 2). A similar conclusion was made by Ladha et al. [13] in their work, where they emphasized the management of N to increase its uptake efficiency. Nitrogen use efficiency (NUE) and agronomic efficiency (AEN) were evaluated by Srivastava et al. [61] under different N rates (0 ($N_0$), 75 ($N_{75}$), 100 ($N_{100}$) and 125 ($N_{125}$) kg ha$^{-1}$ and 0 ($N_0$), 60 ($N_{60}$), 80 ($N_{80}$) and 100 ($N_{100}$) kg ha$^{-1}$)) and sowing scenarios in maize. They concluded that the under rainfed conditions, an N rate from $N_0$ to $N_{100}$ results in decreased NUE and AEN. Today, wheat cultivars require a higher input of N, but this results in a risk of environmental pollution. Therefore, the management of N is very important. Guarda et al. [62] investigated the impacts of different N rates ($N_0$, $N_{80}$, $N_{160}$, kg ha$^{-1}$) on wheat yield quality and NUE. The results showed that the management of N resulted in improved plant N uptake, NUE, and grain quality. Nitrogen uptake efficiency is the measure of how much N is taken up by the wheat crop. Raun and Johnson [63] reported that to increase NUE, its uptake must be enhanced. Raun et al. [64] concluded that N fertilization helps to improve NUE in wheat. The results of the current study showed that NUpE was affected by N treatments during both the years at three study sites. Rahimizadeh et al. [44] found decreased N uptake efficiency under increased N rates, which might be due to more N losses.

Agronomic traits showed significant changes in the present findings for all treatments, and it has been reported that dry matter and grain yield of the plants could be improved by N fertilization [64]. N has shown a positive influence on the growth and development of crop under water stress by sustaining metabolic activities. However, N fertilization should be matched with the crop demand as, in our case, higher agronomic traits were observed for split treatments compared to a full application at the time of sowing. Generally, farmers apply N at the time of sowing or at the earlier growth stages of the crop, which results in the maximum loss of N and lowered crop dry matter and yield. Thus, an integrated soil-crop system management strategy could help to improve grain yield and NUE as proposed by Meng et al. [65]. Hawkesford et al. [66] reported that 33% of applied N fertilizer is recovered in the harvested grain. Thus, 67% of N is lost, which could be a major source of pollutants and should be a major target for crop improvement. Therefore, agronomic management and crop breeding traits could help to improve NUE. Furthermore, optimizing the N application rate could be a good option to minimize N losses and increase crop yield [67]. The relationship of grain yield with crop physiological traits showed that stomatal conductance could be the major determinant of the grain yield. Since global temperature and frequent occurrence of drought could maximize N losses, new avenues for improving crop productivity must thus be exploited [68–72].

## 5. Conclusions

Crop physiological traits, total N, and N efficiencies have a significant strong relationship with crop dry matter and grain yield. Since crop yield is the most commonly used indicator, it needs to be improved by considering the interaction of N addition with crop physiology, such as stomatal conductance, net photosynthetic rate, and transpiration rate. The optimum N application rate and its timings could help to harvest real benefits of N as in our findings split dosage resulted in optimum physiological traits, i.e., $g_s$, $A_n$ and E. Improvement in stomatal conductance through the management of N under stress could be one of the key determinants of crop yield as it balances the crop $CO_2$ uptake and water loss. Similarly, a split application of N resulted in higher N efficiencies, i.e., NUE, nitrogen uptake efficiency, and nitrogen utilization efficiency at all sites for both years. Thus, the idea of

Hawkesford et al. [66] to recover maximum applied N fertilizer is possible through its split application during different stages of the crop. Furthermore, a split application of N resulted in the maximum agronomic traits, and a significant combined strong relationship was obtained between grain yield and crop physiological parameters. The results showed that $T_6$ = split $N_{100}$ could be used to get optimal returns from N. However, in the future, we will be further using the quadratic plateau model approach to build the relationship between N methods and rates. This will ultimately help us to optimize crop physiological traits and grain yield under different sets of N scenarios at these variable field sites.

**Author Contributions:** Experiments, field data collection, and analysis were performed by U.Q. and were part of her Ph.D. thesis under the supervision of M.A. (Mukhtar Ahmed) as a major supervisor, while guided and helped by the F.-u.-H. and M.A. (Muhammad Akmal) Statistical analysis was performed by U.Q. and M.A. (Mukhtar Ahmed). The manuscript was written by U.Q. and M.A. (Mukhtar Ahmed).

**Funding:** This research received no external funding.

**Acknowledgments:** The present work is part of the Ph.D. research work of the first author, and she was supported by the Higher Education Commission (HEC) Pakistan to visit Washington State University (WSU), Pullman USA under the International Research Support Initiative Programme (IRSIP).

**Conflicts of Interest:** The authors declare no conflicts of interest.

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
