# Peer review of "Impact of Nitrogen Addition on Physiological, Crop Total Nitrogen, Efficiencies and Agronomic Traits of the Wheat Crop under Rainfed Conditions"

_sustainability, doi:10.3390/su11226486_

Round 1
Reviewer 1 Report
I am writing to you regarding manuscript # sustainability-626681 entitled "Sustaining Wheat Physiology, biomass and yield through Nitrogen Management Under Contrasting Environmental Conditions ". The author addresses the nitrogen uptake and assimilation in with drought stress. The paper is clear and well written and the experimental design sound. Though there are some typological error, they are mentioned throughout the manuscript. I have few suggestions for the improvement of manuscript as follows:
Â
Introduction: The authors need to revised introduction. Much detailed, latter part of introduction is looking like discussion. Introduction should reflect in concise: background, problem, hypothesis answered in research. Title should be revised. In the discussion section, author should explain, why split application had more pronounced effect in drought conditions. The discussion should include more in-depth explanation of the mechanisms underlying the observed results and explain the implication the results have. In the present version, the discussion is primarily descriptive and comparative with previous literature
Author Response
Author's Reply to the Review Report (Reviewer 1)
Â
Review Report Form
Open Review
(x)Â I would not like to sign my review reportÂ
( )Â I would like to sign my review reportÂ
English language and style
( )Â Extensive editing of English language and style requiredÂ
( )Â Moderate English changes requiredÂ
(x)Â English language and style are fine/minor spell check requiredÂ
( )Â I don't feel qualified to judge about the English language and styleÂ
| Â |
Â
| Â |
Yes |
Can be improved |
Must be improved |
Not applicable |
|
Does the introduction provide sufficient background and include all relevant references? |
( ) |
( ) |
(x) |
( ) |
|
Is the research design appropriate? |
(x) |
( ) |
( ) |
( ) |
|
Are the methods adequately described? |
(x) |
( ) |
( ) |
( ) |
|
Are the results clearly presented? |
(x) |
( ) |
( ) |
( ) |
|
Are the conclusions supported by the results? |
( ) |
( ) |
( ) |
( ) |
Â
Comments and Suggestions for Authors
I am writing to you regarding manuscript # sustainability-626681 entitled "Sustaining Wheat Physiology, biomass and yield through Nitrogen Management Under Contrasting Environmental Conditions ". The author addresses the nitrogen uptake and assimilation in with drought stress. The paper is clear and well written and the experimental design sound. Though there are some typological error, they are mentioned throughout the manuscript. I have few suggestions for the improvement of manuscript as follows:
Introduction: The authors need to revised introduction. Much detailed, latter part of introduction is looking like discussion. Introduction should reflect in concise: background, problem, hypothesis answered in research. Title should be revised.In the discussion section, author should explain, why split application had more pronounced effect in drought conditions. The discussion should include more in-depth explanation of the mechanisms underlying the observed results and explain the implication the results have. In the present version, the discussion is primarily descriptive and comparative with previous literatureÂ
Â
Â
Reply to the Comments:
Suggestions incorporated as suggested by the valuable reviewer. Background, problem and hypothesis have been added in the introduction section. Some of the latter part of the introduction is deleted as well as reworded as reviewer was saying it look like discussion. Title has been changed and now new title isImpact of Nitrogen addition on physiological, crop total nitrogen, nitrogen efficiencies and agronomic traits of the wheat crop under rainfed conditions
Â
Discussion section has been revised as suggested by the reviewer.Â
Â

Reviewer 2 Report
Qadeer et al., reported a study regarding the nitrogen management on physiological, biomass and yield of wheat under contrasting environmental conditions.
Line 23. 2014-25
Lines 57-60. In my experience this statement is partially true, however, only in the first time of adaptation. Please add a reference.
Line 61 add a reference
When the name of a crop appears for the first time in the manuscript the authors should report the corresponding scientific name
Line 446 the authors should report more information on the used cultivar such as the name of the company that commercialize the cultivar, the characteristics of the cultivar, the main use (bred, pasta), the main tolerance/resistance to abiotic and biotic stresses
Authors should report more information regarding soil type (physical and chemical parameters) for each of the investigated sites, the agronomic management in term of soil tillage and crop rotation and the plant density in term of seed per m2.
Why the authors did not measure water use efficiency? Is just a ratio between An and E, and this parameter is essential for this kind of study
Authors should add information on the sustainability of the investigated treatments, for example an analysis of the carbon footprint or at least an analysis of the marginal net return. Have a look at https://doi.org/10.1080/14620316.2019.1577186 and https://doi.org/10.1016/j.eja.2011.06.004
Line 492 biological yield (grain + straw)? I suggest to clarify that in the manuscript
Lines 503-504. I suggest to compare the means using Tukey test as post-hoc test end according revise the result section
For this kind of study is really strange that the authors did not report any information on the soil water content during the crop stage. Soil water content and nitrogen availability are strongly correlated. For example, if the soil water content is too low, farmers can apply nitrogen, but plants are not able to use this nitrogen due to the absence of sufficient soil water content. Have a look to recent works: Field Crops Research 241, 107559 https://doi.org/10.1016/j.fcr.2019.107559 and European Journal of Agronomy 106, 1-11 https://doi.org/10.1016/j.eja.2019.03.002
Author Response
Author's Reply to the Review Report (Reviewer 2)
Open Review
(x)Â I would not like to sign my review reportÂ
( )Â I would like to sign my review reportÂ
English language and style
( )Â Extensive editing of English language and style requiredÂ
( )Â Moderate English changes requiredÂ
(x)Â English language and style are fine/minor spell check requiredÂ
( )Â I don't feel qualified to judge about the English language and styleÂ
| Â |
Â
| Â |
Yes |
Can be improved |
Must be improved |
Not applicable |
|
Does the introduction provide sufficient background and include all relevant references? |
(x) |
( ) |
( ) |
( ) |
|
Is the research design appropriate? |
( ) |
(x) |
( ) |
( ) |
|
Are the methods adequately described? |
( ) |
(x) |
( ) |
( ) |
|
Are the results clearly presented? |
( ) |
(x) |
( ) |
( ) |
|
Are the conclusions supported by the results? |
( ) |
(x) |
( ) |
( ) |
Comments and Suggestions for Authors
Qadeer et al., reported a study regarding the nitrogen management on physiological, biomass and yield of wheat under contrasting environmental conditions.
Comment#1
Line 23. 2014-25
Reply to the Comment:
Corrected as suggested.
Comment#2
Lines 57-60. In my experience this statement is partially true, however, only in the first time of adaptation. Please add a reference.
Reply to the Comment:
Reference added as suggested.
Zhang, X.C.; Yu, X.F.; Ma, Y.F. Effect of nitrogen application and elevated CO2 on photosynthetic gas exchange and electron transport in wheat leaves. Photosynthetica 2013, 51, 593-602, doi:10.1007/s11099-013-0059-5.
Comment#3
Line 61 add a reference
Reply to the Comment:
Reference added as suggested.
Comment#4
When the name of a crop appears for the first time in the manuscript the authors should report the corresponding scientific name
Reply to the Comment:
Suggestion incorporated.
Comment#5
Line 446 the authors should report more information on the used cultivar such as the name of the company that commercialize the cultivar, the characteristics of the cultivar, the main use (bred, pasta), the main tolerance/resistance to abiotic and biotic stresses
Reply to the Comment:
Suggestion incorporated. Name of the company have been added.
Comment#6
Authors should report more information regarding soil type (physical and chemical parameters) for each of the investigated sites, the agronomic management in term of soil tillage and crop rotation and the plant density in term of seed per m2.
Reply to the Comment:
Soil information added as suggested. Please see table 5 and 6.
Agronomic management incorporated as suggested by the valuable reviewer.
Â
Comment#7
Why the authors did not measure water use efficiency? Is just a ratio between An and E, and this parameter is essential for this kind of study
Reply to the Comment:
Agreed we have calculated this, but this will be presented in another article separately and that will be only about water use efficiency.
Comment#8
Authors should add information on the sustainability of the investigated treatments, for example an analysis of the carbon footprint or at least an analysis of the marginal net return. Have a look at https://doi.org/10.1080/14620316.2019.1577186 and https://doi.org/10.1016/j.eja.2011.06.004
Reply to the Comments:
Firstly, thanks to the reviewer for providing these very interesting articles and important suggestions. We highly appreciate this. We have calculated the marginal net return but same as above comments will be willing to use this idea as separate article. Similarly, we will calculate carbon footprint for future article as if we add here then this article will be very big.
Comment#9
Line 492 biological yield (grain + straw)? I suggest to clarify that in the manuscript
Reply to the Comment:
Suggestion incorporated.
Comment#10
Lines 503-504. I suggest to compare the means using Tukey test as post-hoc test end according revise the result section
Reply to the Comment:
Suggestion incorporated.
Comment#11
For this kind of study is really strange that the authors did not report any information on the soil water content during the crop stage. Soil water content and nitrogen availability are strongly correlated. For example, if the soil water content is too low, farmers can apply nitrogen, but plants are not able to use this nitrogen due to the absence of sufficient soil water content. Have a look to recent works: Field Crops Research 241, 107559 https://doi.org/10.1016/j.fcr.2019.107559 and European Journal of Agronomy 106, 1-11 https://doi.org/10.1016/j.eja.2019.03.002
Reply to the Comment:
Valuable suggestion incorporated.
Â

Reviewer 3 Report
Hanna Klikocka                                         Lublin, 25.10.2019.
University of Life Sciences
In Lublin, Poland
Â
Â
Paper 626681:
 Sustaining Wheat Physiology, biomass and yield through Nitrogen Management  Under Contrasting Environmental Conditions Â
Â
Â
I would like to kindly inform you that the manuscript submitted for evaluation needs to be corrected. In particular, I provide below what elements of the work the authors should complete or improve. On the manuscript in yellow windows I made comments that also apply to the editorial page.
The publication is interesting and multi-threaded. Deserves to be published in the selected journal. However, it needs improvement. After improvement, I recommend it for printing.
Â
Abstract:
Please add the BBCH scale when describing the vegetation phase of wheat
Materials and Methods.
Please, preferably give in the table the soil type, its granulometric composition, pH of soil and N-total content in the soil, How many seeds were sown per 1 m2, What acid was used to determine total N,
Results – Section 2.1.
Please use the description as done in the following sections.
Table:
Provide explanations for the features examined under the table.
                                                                                                                       Sincerely
Â

Author Response
Author's Reply to the Review Report (Reviewer 3)
Open Review
(x)Â I would not like to sign my review reportÂ
( )Â I would like to sign my review reportÂ
English language and style
(x)Â Extensive editing of English language and style requiredÂ
( )Â Moderate English changes requiredÂ
( )Â English language and style are fine/minor spell check requiredÂ
( )Â I don't feel qualified to judge about the English language and styleÂ
| Â |
Â
| Â |
Yes |
Can be improved |
Must be improved |
Not applicable |
|
Does the introduction provide sufficient background and include all relevant references? |
(x) |
( ) |
( ) |
( ) |
|
Is the research design appropriate? |
( ) |
(x) |
( ) |
( ) |
|
Are the methods adequately described? |
( ) |
(x) |
( ) |
( ) |
|
Are the results clearly presented? |
( ) |
( ) |
(x) |
( ) |
|
Are the conclusions supported by the results? |
(x) |
( ) |
( ) |
( ) |
Comments and Suggestions for Authors
Hanna Klikocka                                         Lublin, 25.10.2019.
University of Life Sciences
In Lublin, Poland
General Comments
Paper 626681:  Sustaining Wheat Physiology, biomass and yield through Nitrogen Management  Under Contrasting Environmental Conditions Â
I would like to kindly inform you that the manuscript submitted for evaluation needs to be corrected. In particular, I provide below what elements of the work the authors should complete or improve. On the manuscript in yellow windows I made comments that also apply to the editorial page. The publication is interesting and multi-threaded. Deserves to be published in the selected journal. However, it needs improvement. After improvement, I recommend it for printing.
 Reply to the General Comments: Correction made as suggested by the valuable reviewer.
Â
Comment#1
Abstract:
Please add the BBCH scale when describing the vegetation phase of wheat
Reply to the Comment:
Suggestions incorporated as BBCH scale of wheat.
Comment#2
Materials and Methods.
Please, preferably give in the table the soil type, its granulometric composition, pH of soil and N-total content in the soil, How many seeds were sown per 1 m2, What acid was used to determine total N,
Reply to the Comment:
Details added as suggested by the valuable reviewer. Kindly see table 5 and 6 please.
Comment#3
Results – Section 2.1.
Please use the description as done in the following sections.
Reply to the Comment: Suggestion incorporated.
Table:
Provide explanations for the features examined under the table.
    Reply to the Comment: Suggestion incorporated.
REPLY TO THE COMMENTS IN THE MAIN PDF FILE
Abstract
Comment#1: Add digit (Line#23)
Reply to the Comment: Added
Â
Comment#2:apply BBCH scale (Line#28)
Reply to the Comment: Applied.
Â
Comment#3: Small letter (Line#31)
Reply to the Comment: Converted as directed
Â
Introduction
Comment#1: No index (Line#93,97,99,104,125)
Reply to the Comment: Index corrected everywhere as suggested
Comment#2: Please start with a new paragraph
Reply to the Comment: Needful done as suggested.
Â
Results
Comment#1: Please use the description as done in the following sections.
Reply to the Comment: Suggestions incorporated as directed
Â
Discussions
Comment#1: No index (Line#338)
Reply to the Comment: Index corrected everywhere as suggested
Â
Materials and Methods
Comment#1: Please, preferably give in the table the soil type, its granulometric composition, pH and N-total content.
Reply to the Comment: Required information has been added.
Â
Comment#2: Add BBCH scale
Reply to the Comment: BBCH scale added.
Â
Comment#3: How many seeds were sown per 1 m2
Reply to the Comment: Seed rate used is added.
Â
Comment#4:Better BBCH scale
Reply to the Comment: Suggestion incorporated.
Â
Comment#4: What acid was used to determine total N
Reply to the Comment: Concentrated Sulfuric acid was used to determine total N. Further detail has been added.
Â
Â
Conclusions
Comment#1: No index (Line#518)
Reply to the Comment: Index corrected as suggested
Â
Tables
Comment#1: Provide explanations for the features examined under the table
Reply to the Comment: Suggestion incorporated.
